# Evaluation of Viscosity Changes and Rheological Properties of Diutan Gum, Xanthan Gum, and Scleroglucan in Extreme Reservoirs

**DOI:** 10.3390/polym15214338

**Published:** 2023-11-06

**Authors:** Xin Gao, Lixin Huang, Jianlong Xiu, Lina Yi, Yongheng Zhao

**Affiliations:** 1School of Engineering Science, University of Chinese Academy of Sciences, Beijing 100049, China; gaoxin211@mails.ucas.ac.cn (X.G.); zhaoyongheng21@mails.ucas.ac.cn (Y.Z.); 2Institute of Porous Flow and Fluid Mechanics, Chinese Academy of Sciences, Langfang 065007, China; yisheng_218@163.com; 3State Key Laboratory of Enhanced Oil Recovery, PetroChina Research Institute of Petroleum Exploration & Development, Beijing 100083, China; yilina69@petrochina.com.cn

**Keywords:** diutan gum, xanthan gum, scleroglucan, rheological behavior, extreme reservoir, long-term stability

## Abstract

The chemically synthesized polymer polyacrylamide (HPAM) has achieved excellent oil displacement in conventional reservoirs, but its oil displacement is poor in extreme reservoir environments. To develop a biopolymer oil flooding agent suitable for extreme reservoir conditions, the viscosity changes and rheological properties of three biopolymers, diutan gum, xanthan gum, and scleroglucan, were studied under extreme reservoir conditions (high salt, high temperature, strong acid, and alkali), and the effects of temperature, mineralization, pH, and other factors on their viscosities and long-term stability were analyzed and compared. The results show that the three biopolymers had the best viscosity-increasing ability at temperatures of 90 °C and below. The viscosity of the three biopolymers was 80.94 mPa·s, 11.57 mPa·s, and 59.83 mPa·s, respectively, when the concentration was 1500 mg/L and the salinity 220 g/L. At the shear rate of 250 s^−1^, 100 °C~140 °C, scleroglucan had the best viscosification. At 140 °C, the solution viscosity was 19.74 mPa·s, and the retention rate could reach 118.27%. The results of the long-term stability study showed that the solution viscosity of scleroglucan with a mineralization level of 220 mg/L was 89.54% viscosity retention in 40 days, and the diutan gum could be stabilized for 10 days, with the viscosity maintained at 90 mPa·s. All three biopolymers were highly acid- and alkali-resistant, with viscosity variations of less than 15% in the pH3~10 range. Rheological tests showed that the unique double-helix structure of diutan gum and the rigid triple-helix structure of scleroglucan caused them to have better viscoelastic properties than xanthan gum. Therefore, these two biopolymers, diutan gum, and scleroglucan, have the potential for extreme reservoir oil displacement applications. It is recommended to use diutan gum for oil displacement in reservoirs up to 90 °C and scleroglucan for oil displacement in reservoirs between 100 °C and 140 °C.

## 1. Introduction

Microbial polysaccharide is a substance produced by microorganisms during growth and metabolism. It is a novel biopolymer possessing the physical and chemical properties of thickening, emulsification, stability, and gelation, along with biocompatibility, sustainability, and environmentally friendly features. This renders it highly versatile and useful in a variety of areas, including in food, petroleum, cement, textiles, and cosmetics [1,2,3,4]. In the petroleum industry, biopolymers are widely used in oilfield drilling fluids, fracturing fluids, and profile control and water plugging [1]. Drilling fluid is a multifaceted mixture consisting of solids, liquids, and chemical agents. Xanthan gum, guar gum, starch, and cellulose derivatives are commonly used biological viscosifiers in drilling fluids due to their good viscosity-increasing properties. However, under extreme high-temperature and high-salt circumstances, the fluid’s weaknesses, including poor thermal stability, weak shear capacity, and reduced viscosity-increasing ability, become more evident [1,5,6,7,8]. Hydraulic fracturing technology is a commonly employed method for modifying reservoirs. The effectiveness of the fracturing fluid is crucial to the success of the operation. Biopolymers, such as guar gum, are added to the fracturing fluid as thickeners with retarding properties. Enhancing the temperature resistance of guar gum is achievable by modifying its molecular chain and grafting it with rigid agents. To broaden its application scope, common cross-linking agents include borate, Ti^4+^, Zr^4+^, and Al^3+^ [9,10]. As oilfield water plugging agents, biopolymers not only improve the water absorption profile and displacement effect of water injection wells but also expand the effective layers and directions of oil wells, thereby overall improving oil recovery. Biopolymers currently used for water plugging and profile control include modified starch, modified cellulose, guar gum, xanthan gum, chitosan, etc. However, the majority of biopolymer plugging agents are currently still undergoing laboratory research and are only being produced in small batches [1].

Currently, the majority of oil fields in China have entered the tertiary oil recovery stage [11,12], among which, polymer flooding is widely used because it produces more oil than other chemically enhanced oil recovery methods and is environmentally friendly. The reservoir environment becomes increasingly hostile as the depth of reservoir formation deepens, the concentration of formed water ions increases, and oil displacement technologies to alter the acidity and alkalinity of the reservoir proliferate, such as CO_2_ flooding, alkali flooding, and ternary composite flooding. At present, polymer flooding is being developed from use in medium- and low-temperature oil reservoirs to application in high-temperature and high-salt oil reservoirs. However, the conditions of high-temperature and high-salt oil reservoirs are complex; the oil displacement mechanism needs to be deepened, and the compatibility of the oil displacement system is poor. There is an urgent need for temperature and salt resistance and adaptability to harsh reservoir environments in polymer oil-displacing agents [13]. The polymer oil displacement system commonly used in oilfields is based predominately on partially hydrolyzed polyacrylamide (HPAM) and its modified products [14,15,16,17,18]. This system has an excellent oil displacement effect in low- and medium-temperature and low- and medium-mineralization reservoirs, but is subject to the influence of temperature, mineralization, and acidity and alkalinity, and the environment is prone to pollution. The use of biopolymers to improve the apparent viscosity of the oil displacement system and to expand the efficiency of the wave and the effect of the system is not weaker compared with HPAM, and they can be degraded in a safe and environmentally friendly way.

In recent years, numerous oil fields have implemented microbial-enhanced oil recovery technology (MEOR) to meet the requirements of green and efficient development. This technology has received attention owing to its eco-friendly attributes, superior applicability, and cost-effectiveness [19,20,21,22,23]. Zhou Hongtao [13] explored the feasibility of diutan gum, welan gum, and xanthan gum as new polymers for oil repulsion; Zhang Xifeng [24] investigated the static and dynamic rheological properties of a welan gum aqueous solution at a formation temperature of 40 °C and the factors affecting them; Karl-Jan [25] investigated the rheology of schizophyllan, scleroglucan, guar gum, and xanthan gum in brines at concentrations ranging from 10 to 2300 mg/L, temperature levels ranging from 25 to 70 °C, and total dissolved solids concentrations of 30,100 mg/L and 69,100 mg/L; and Lai Nanjun [26] investigated the rheological properties and the recovery-enhancing ability of diutan gum, xanthan gum, and konjac gum at a temperature of 130 °C and a mineralization level of 223.07 mg/L. However, less research has been carried out on the viscosity enhancement, rheological properties, and long-term stability of the three biopolymers, namely, diutan gum, xanthan gum, and scleroglucan, under extreme reservoir conditions.

The main objective of this paper is to explicitly determine the viscosity changes and rheological properties, as well as the oil drive potential, of diutan gum, xanthan gum, and scleroglucan solutions under extreme reservoir conditions (high salt, high temperature, strong acid, and alkali). Therefore, the viscosity concentration relationship, steady-state rheological shear properties, linear viscoelastic region LVR and oscillatory frequency scanning, temperature resistance, salt resistance, acid and alkali resistance, and long-term stability in extreme environments were investigated in detail. These properties were compared with each other to provide technical support for the construction of a polymer oil displacement system.

## 2. Materials and Methods

### 2.1. Experimental Materials

Diutan gum and scleroglucan concentrate were supplied by Hebei Xinhe Bio-Chemical Co., LTD. (Xinhe County, China), while xanthan gum was supplied by Meihua Bio-Technology Group Co., LTD. (Langfang, China). The diutan gum was a light-yellow colloid (effective content of 2%, molecular weight 5.2 × 10^6^ Da); the scleroglucan was a milky-white colloid (effective content 2%, molecular weight 1.3 × 10^5^~6 × 10^6^ Da); and the xanthan gum was a light-yellow powder solid (effective content 90%, molecular weight 2 × 10^6^ Da).

The biopolymer solution was prepared with six kinds of simulated brine with different salinity levels. The 220 g/L simulated brine was the mother liquor, and the remaining five simulated brines were diluted from the mother liquor. Their compositions are shown in Table 1. The simulated brine preparation water was distilled water, and sodium chloride, calcium chloride, magnesium chloride hexahydrate, sodium hydroxide, and hydrochloric acid reagent grades were purchased from China Sinopharm Chemical Reagent Co., Ltd. Unless otherwise specified, the biopolymer solutions were prepared using 6 g /L simulated saline.

### 2.2. Solution Preparation

#### 2.2.1. Preparation Method for Diutan Gum and Scleroglucan

At room temperature (25 °C), an appropriate amount of biopolymer concentrate was placed in a beaker, a small amount of simulated brine was added, and the solution was stirred at 1000 r/min for 10 min. The remaining simulated brine was added to finalize the solution and it was stirred for another 3 h until the biopolymer was sufficiently dissolved to obtain the biopolymer solution with the target mass concentration. The preparation water not mentioned in the test experiments was simulated brine with a total mineralization of 6 g/L.

#### 2.2.2. Preparation Method for Xanthan Gum

The simulated brine was first made to form a vortex at 800 r/min, and then the xanthan gum powder was added slowly along the wall of the vortex. After all the xanthan gum was added, it was stirred at 800 r/min for 30 min, and then the speed was lowered to 600 r/min for continuous stirring until dissolution was complete.

Before testing, the solution was left at room temperature for 12 h to fully swell and mature, allowing internal air bubbles to be eliminated to prevent interference with later performance measurement research.

### 2.3. Experimental Instruments

The Thermo Fisher Scientific Haake rotation rheometer MARS 60, equipped with two test systems, a CC41 DG/Ti-02210522 double-cylinder test system and PZ DG 38 Ti high- temperature sealed test system, was from Thermo Fisher Company, Germany; the Brookfield viscometer was from the Brookfield Company of the United States; the electric blast drying oven was from Shanghai Yiheng Scientific Instrument Co., Ltd., Shanghai, China; the Electric constant- temperature water bath, was from Shanghai Boxun Medical Biological Instrument Corp, Shanghai, China; the pH meter was from Shanghai INESA Scientific Instrument Co, Ltd., Shanghai, China; the cantilever electric stirrer instrument was from Shanghai Lichenbangxi Instrument Technology Co., Ltd., Shanghai, China; and the electronic balance was from Shanghai Zhuojing Electronic Technology Co., Ltd., Shanghai, China.

### 2.4. Experimental Test Methods

#### 2.4.1. Viscosity Increase Performance Test

The apparent viscosities of the three biopolymer solutions at different concentrations were determined at 20 °C and 90 °C to examine the relationship between apparent viscosity and solution concentration. Three biopolymer solutions with a concentration of 1500 mg/L were prepared according to the method described in Section 2.1 using 6 g/L of simulated brine under two conditions to determine the apparent viscosity of the three biopolymer solutions as a function of temperature: (1) in a temperature range of 20–90 °C with a shear rate of 7.34 s^−1^ using a CC41 DG/Ti–02210522 double-cylinder test system; and (2) in a temperature range of 70–150 °C with a shear rate of 250 s^−1^ using a PZ DG 38 Ti high-temperature confinement test system. The six simulated brines above, with different mineralizations, were used to formulate biopolymer solutions at a concentration of 1500 mg/L according to the method described in Section 2.1, and the apparent viscosity changes of biopolymer solutions with different mineralizations at different temperatures were determined. Three biopolymer solutions with a concentration of 1500 mg/L were prepared using 6 g/L of simulated brine, and the viscosity changes with the pH of the three biopolymer solutions were determined in the pH range of 3–10, at a temperature of 90 °C and a shear rate of 7.34 s^−1^.

#### 2.4.2. Rheological Properties Tests

All rheological properties, including steady-state rheological shear, oscillation amplitude sweep, and oscillation frequency sweep tests, were performed on a Haake rotational rheometer Haake MARS 60, using a CC41 DG/Ti-02210522 double-cylinder test system and a PZ DG 38 Ti high-temperature closed testing system. All the tests were repeated at least three times, and the relative deviation was less than 0.05%.

Steady-state rheological shear performance test.

The steady-state rheology of three biopolymer solutions was tested to examine the relationship between apparent viscosity and shear rate. The shear rate range was 0.01~1000 s^–1^, the temperature was 90 °C, and the temperature deviation was controlled at ±0.1 °C; the polymer solution mass concentration was 1500 mg /L. The measured steady-state rheological curve was fitted using the power-law function (1):(1)$$\eta_{v}=k\cdot \gamma^{n-1}$$

In the formula, $\eta_{v}$ is the apparent viscosity; k is the consistency factor; n is the power-law index, which is dimensionless; and γ is the shear rate in s ^–1^.

2.Linear viscoelastic zone LVR and oscillation frequency scanning tests.

First, strain amplitude scanning was performed on the three biopolymer solutions to determine the linear viscoelastic region LVR of the solution. The concentration of the prepared solution was 1500 mg /L, the oscillation frequency was fixed at 1 Hz, the temperature was 90 °C, and the strain was from 0.01% to 100%; after determining the linear viscoelastic zone L VR of the solution, the solution was then subjected to an oscillation frequency scan test, in which the shear stress was fixed, and the frequency scan was performed in the range of 0.01–10.00 Hz at 90 °C.

#### 2.4.3. Long-Term Stability Performance Test

A 1500 mg/L quantity of diutan gum and 400 mL of scleroglucan biopolymer solution, respectively, were added to 20 high-temperature-resistant glass bottles with a volume of 12 mL, into which nitrogen was injected and sealed to remove the dissolved oxygen in the solution and prevent oxygen oxidation of the solution from interfering with the apparent viscosity data. Then the glass bottles containing the biopolymer solution were placed into a 100 °C constant temperature oven to age and taken out at intervals to determine the apparent viscosity.

## 3. Results

### 3.1. Comparison of the Viscosity-Increasing Properties of Three Biopolymers

#### 3.1.1. Viscosity Concentration Relationships of Three Biopolymers

The apparent viscosities of the three biopolymer solutions were measured at a shear rate of 7.34 s^−1^ and temperatures of 20 °C and 90 °C. As can be seen in Figure 1, the apparent viscosities of all three biopolymer solutions increased with the increase in solution concentration, but the apparent viscosities of the biopolymer solutions increased to different extents at different temperatures for different concentrations. For example, at 90 °C, all three biopolymer solutions had a concentration of 2500 mg/L, and the apparent viscosities were 223.8 mPa·s for diutan gum, 169.45 mPa·s for scleroglucan, and 121.7 mPa·s for xanthan gum; the apparent viscosities of the diutan gum and scleroglucan solution were nearly 1.84 times and 1.39 times that of xanthan gum, respectively. Figure 2 is the change curve of the biopolymer solution concentration with apparent viscosity at 90 °C. To ensure the solution has the same apparent viscosity (100 mPa·s), it is necessary to add 1414 mg/L of diutan gum or 1725 mg/L of scleroglucan, while for xanthan gum, a higher concentration is needed and, under the same conditions, is necessary to achieve the same apparent viscosity, which is about 2140 mg/L.

The diutan gum solution showed better viscosity thickening than the xanthan gum and scleroglucan solutions. This was due to the high molecular weight of diutan gum, the increase in the entanglement of the polymer chains at high concentrations, the corresponding increase in apparent viscosity, and the formation of agglomerates at low shear rates (7.34 s^−1^) by entangling the stretched molecules with each other, resulting in a greater resistance to fluid flow and leading to a better viscosity enhancement effect. This is similar to the findings of Xu [2], Holzwarth [27], and G.P. Mota [28].

#### 3.1.2. Temperature Resistance Performance Test

The solutions of diutan gum, xanthan gum, and scleroglucan were prepared at a concentration of 1500 mg/L, and the apparent viscosity versus temperature curves of the three biopolymer solutions were measured at a low shear rate (7.34 s^−1^) and a temperature range of 20–90 °C, as shown in Figure 3. It can be seen that all three polymer solutions exhibited thermal thinning behavior [29]: the apparent viscosity decreased to varying degrees with increasing temperature, with the apparent viscosity of diutan gum and scleroglucan solutions decreasing to a lesser extent, from 105.6 mPa·s and 96.28 mPa·s at the initial 20 °C to 94.17 mPa·s and 85.21 mPa·s at 90 °C, whereas the apparent viscosity of xanthan gum solution decreased further, from 78.12 mPa·s at the initial 20 °C to 47.42 mPa·s at 90 °C.

Figure 4 shows the apparent viscosity retention φ (φ, the ratio of apparent viscosity at different temperatures to the initial apparent viscosity at 20 °C) of the three biopolymer solutions at temperatures ranging from 20 to 90 °C and at a shear rate of 7.34 s^−1^. The apparent viscosity retention φ of diutan gum and scleroglucan solution changed slightly with the increase in temperature, while the retention φ of xanthan gum solution gradually decreased and the decrease was larger. Additionally, at the same temperature, the apparent viscosity retention φ of diutan gum and scleroglucan solution was always higher than that of xanthan gum, and the apparent viscosity retention φ of the diutan gum and scleroglucan solution were 89.18% and 88.50%, respectively, while that of the xanthan gum solution was only 60.70% at the temperature of 90 °C.

Under the above experimental conditions, the CC41 DG/Ti-02210522 two-barrel test system of the Haake Rheometer was used, and the temperature limit of this system was 90 °C. For the higher temperature range, a Haake Rheometer PZ DG 38 Ti high-temperature confinement test system was used. However, this test system is only suitable for high shear rates (>170 s^−1^), so the apparent viscosity of the three biopolymer solutions was tested as a function of temperature in the temperature range of 70–150 °C and at a shear rate of 250 s^−1^.

As shown in Figure 5 and Figure 6, between 70 and 150 °C, it can be seen that the apparent viscosity of the diutan gum and xanthan gum solutions decreased significantly with increasing temperature, while the apparent viscosity of the scleroglucan solution, on the other hand, increased slightly. Compared with 70 °C, at 150 °C, the apparent viscosity retention rates φ of diutan gum and xanthan gum solutions were only 44.83% and 19.3%, respectively, while the scleroglucan solution had an apparent 118.27% viscosity retention rate, and was still able to maintain a higher apparent viscosity under the ultra-high temperature.

It is clear that diutan gum and scleroglucan are less sensitive to temperature than xanthan gum. Within 20–70 °C, the apparent viscosity value and the degree of temperature stability of the solutions of diutan gum and scleroglucan were always better than those of xanthan gum, and as can be seen in Figure 5 and Figure 6, at 100 °C, the apparent viscosity of diutan gum was 17.39 mPa·s and the apparent viscosity retention rate was 89.00%, but at 110 °C, its apparent viscosity retention rate was only 67.40%, which was a large decrease. It can be concluded that the diutan gum was tolerant to the 100 °C high temperature. On the other hand, the xanthan gum solution, with apparent viscosity retention rates of 74.80% and 57.32% at 100 °C and 110 °C, respectively, had poor temperature resistance. Meanwhile, the apparent viscosity of scleroglucan increased with temperature in the range of 70–150 °C, proving that its temperature resistance was superior.

Commonly, the entanglement of biological macromolecules and the strength of polymer networks are predominately formed by van der Waals forces and hydrogen bonds [30]. An increase in temperature accelerates the movement of molecules and distorts a large number of hydrogen bonds, leading to a gradual loss of the second nearest-neighbor spatial correlation and weakening the van der Waals forces and hydrogen bonds [2,31]. At high temperatures, water molecules in the molecular chain escape easily due to the weakening of hydrogen bonds and the thermal motion of the molecules.

In the xanthan gum solution, the reticular structure existed at 20 °C but had been destroyed by 90 °C. Water molecules adhering to the edges of the double helix detached from the molecular chain, leading to a decrease in water retention. Meanwhile, xanthan gum molecules underwent a conformational transition from an ordered double-helical structure to a disordered helical structure in the process of temperature increase, weakening the association between molecular chains, and the apparent viscosity gradually decreases [2,13]. Due to the rigidity of the molecular chain, the double-helix structure of diutan gum could still maintain its conformation well at higher temperatures (≤100 °C), and a large number of water molecules were still tightly trapped in the double-helix structure due to the strong internal force, so it had strong temperature resistance [2,28,29]. The molecular chain of scleroglucan has a triple-helix structure in aqueous solution, and this structure, due to the stabilization of intramolecular and intermolecular hydrogen bonding, results in scleroglucan exhibiting a strong stability over a wide range of temperatures (≤150 °C) [32,33,34,35], which is also approximate to the findings of Kalpakci [36] et al. Therefore, it can be concluded that diutan gum and scleroglucan solutions have better temperature resistance compared with xanthan gum solutions.

#### 3.1.3. Comparison of Salt Resistance of Three Biopolymers

To investigate the salt resistance of the three biopolymer solutions at different temperatures, six simulated brines with different mineralizations were prepared for the formulation of 1500 mg/L biopolymer solutions, and the changes in apparent viscosity with mineralization were determined at 20 °C, 55 °C, and 90 °C, as shown in Figure 7, Figure 8 and Figure 9, with a shear rate of 7.34 s^−1^. Moreover, Figure 10 shows the curve of viscosity retention φ of the biopolymer solution as a function of mineralization at 90 °C.

It can be seen that at these three temperatures, when the mineralization of the biopolymer solution was increased from 6 g/L to 220 g/L, the apparent viscosity of the diutan gum solution essentially remained high and unchanged, and the viscosity retention φ was still maintained at 90.12% at 90 °C. The apparent viscosity of the scleroglucan solution decreased slightly, with a viscosity retention φ of 70.04% at 90 °C. The apparent viscosity of the xanthan gum solution decreased relatively significantly from 47.42 mPa·s to 11.57 mPa·s, with a viscosity retention φ of only 24.40% at 90 °C.

Diutan gum and xanthan gum are polyelectrolytes, which are polyanions in aqueous solution [29,37,38]. For anionic polyelectrolytes, cations (Na^+^, Ca^2+^, and Mg^2+^) in the simulated brine play two main negative roles: (1) shielding electrostatic repulsion between charged groups of macromolecules, enabling macromolecules to adopt more compact conformations; and (2) densifying the hydration layer around the biopolymer molecules, thus reducing the apparent viscosity of diutan gum and xanthan gum solutions [2,29,39].

The side chains of the diutan gum molecules are entangled inside the main chains, and the shielding effect of inorganic cations is limited, which has no effect on the water molecules wrapped in the core of the double helix; on the other hand, xanthan gum has a disordered conformation in pure water or low-concentration simulated brine, and as the mineralization level increases, xanthan gum starts to show an ordered conformation due to the charge shielding effect. Meanwhile, according to Long Xu‘s study [2], the presence of inorganic salts did not change the viscoelastic properties of the diutan gum, and the elastic component was still dominant, while the dynamic modulus of xanthan gum was reduced, which indicated that the diutan gum solution had a better resistance to high salt.

Scleroglucan is a non-ionic water-soluble biopolymer produced by *Sclerotium rolfsii* [40]. Due to its peculiar rigid triple-helix rod structure, the molecules in scleroglucan solutions are virtually independent of the ionic environment, and thus, are insensitive to solution salinity [29,41].

#### 3.1.4. Comparisons of the Acid and Alkali Resistance of Three Biopolymers

Most of reservoirs are in a neutral environment, but gas flooding in tertiary oil recovery, especially CO_2_ flooding, changes the reservoir’s acid-base environment, putting the reservoir in an acidic environment, while some fields use alkali flooding or ternary composite flooding, bringing the reservoir into an alkaline environment. Therefore, it is essential to study the effect of pH on the apparent viscosity of the biopolymer solutions. Three biopolymer solutions were prepared with simulated brine with a salinity of 6 g/L at different pH levels.

The apparent viscosity of the biopolymer solutions changed with pH at 90 °C, as shown in Figure 11. The apparent viscosities of the three biopolymer solutions were almost unchanged in the pH range of 5–7, and the apparent viscosities of the three biopolymer solutions decreased to different degrees in a more acidic environment. The xanthan gum solution decreased to a large extent (19.05%), followed by diutan gum and scleroglucan; in an alkaline environment, the apparent viscosity of the scleroglucan solution slightly increased (up 8.34%), the diutan gum solution was stable, and the apparent viscosity of the xanthan gum solution slightly decreased, which was due to the destruction of the xanthan gum part of the intramolecular and intermolecular hydrogen bonding by the strong alkali [13,42]. However, the diutan gum molecules were not destroyed. The stability of scleroglucan’s acid-base resistance is due to its non-ionic properties and molecular rigidity. There are hydroxyl groups in the glucose unit, and the triple-helix structure is stabilized through interchain hydrogen bonds, so its structure will not change due to the acid-base environment [35]. Diutan gum and scleroglucan solution viscosity was not very affected by alkali concentration. Thay can be used in a wide range of alkali concentrations and can be used in oil fields to improve oil recovery by compounding with alkalis.

### 3.2. Rheological Properties of Three Biopolymers

#### 3.2.1. Steady-State Rheological Shear Test

Steady-state rheological shear tests were performed on the biopolymer solutions, and Figure 12 shows the flow curves of the biopolymer solutions at a temperature of 90 °C (with shear rates in the range of 0.01–1000 s^−1^), comparing the apparent viscosity of diutan gum, xanthan gum, and scleroglucan solutions as a function of shear rate. It can be seen that at elevated temperatures, the apparent viscosities of all three biopolymer solutions decreased with increases in shear rate and showed a strong shear thinning behavior, which was an obvious pseudoplastic fluid behavior. At the same time, the apparent viscosities of the three biopolymer solutions did not differ greatly at high shear rates (greater than 100 s^−1^), and the apparent viscosities of the diutan gum and scleroglucan solutions were higher than that of the xanthan gum solution at low shear rates (less than 100 s^−1^). This is due to the fact that this shear-thinning property is closely related to the state of molecular aggregation or dispersion in the shear flow [43]. Biopolymer molecules normally exist as aggregates at low shear rates, but when the shear rate increases, the aggregates gradually dissociate under shear and the individual molecules rearrange themselves along the direction of flow. The result is a decrease in apparent viscosity, and the molecular weights of diutan gum and scleroglucan are larger relative to those of xanthan gum, which can exhibit higher viscosities at low shear rates [3,44].

Table 2 shows the parameters of the power-law model for the apparent viscosity versus the shear rate of diutan gum, xanthan gum, and scleroglucan solutions at 90 °C. The power-law model fits well (R^2^ > 0.99) for the pseudoplastic behavior of xanthan gum, diutan gum, and scleroglucan solutions in the range of shear rates from 0.01 to 1000 s^−1^, and has been used numerous times by other researchers to fit the relationship between the apparent viscosity and shear rate of biopolymer solutions [2,28,39,41,45,46]. As shown in Table 2, the power rate exponent n of all three was less than 1, which indicates pseudoplastic fluid behavior remains, and the consistency exponent k of the diutan gum and scleroglucan solutions was larger than that of the xanthan gum solution, as shown in Figure 12. The apparent viscosity of diutan gum and hard scleroglucan solution was higher compared with that of the xanthan gum solution, and the strengths of thickening performances of the three were as follows: diutan gum > scleroglucan > xanthan gum.

#### 3.2.2. Linear Viscoelastic Zone LVR Test

A dynamic rheological test of the biopolymer solutions is used to better evaluate their structure. The methods commonly used to examine the structure of the solutions are oscillatory amplitude scans and oscillatory frequency scans. Oscillation amplitude scanning is first used to determine the linear viscoelastic region LVR of biopolymer solutions because, in general, the appropriate strain applied by oscillation frequency scanning must be within the linear viscoelastic region [39,47].

A high plateau of elastic modulus G’ independent of shear stress τ or shear strain is considered to be the linear viscoelastic region. The linear viscoelastic regions of the three biopolymer solutions are shown in Figure 13, which shows that a linear viscoelastic region existed for diutan gum, scleroglucan, and xanthan gum, and that the modulus of elasticity, G’, remained virtually constant with increasing shear stress, τ, until it reached the critical stress value, τ_c_ (τ_c_, the stress at the inflection point). The critical stress value τ_c_ reflects the shear resistance of the molecular agglomerates, and the initial molecular agglomerates are destroyed when τ >τ_c_. If the critical stress value τ_c_ of the solution is large, the molecular chains overlap and entangle with each other and can withstand greater shear stress, so they will have excellent viscoelastic properties [39]. The plateau values of the diutan gum and scleroglucan solutions were higher than that of xanthan gum, and their critical stress values τ_c_ were also higher than that of xanthan gum. This indicates that the viscoelasticity of the diutan gum and scleroglucan solutions is more prominent compared with xanthan gum, and they can withstand larger stresses.

#### 3.2.3. Oscillation Frequency Sweep Test

Figure 14 exhibits the dynamic modulus (i.e., storage modulus G′ and loss modulus G″) of diutan gum, xanthan gum, and scleroglucan solutions versus the oscillation frequency at a temperature of 90 °C and 0.1 Pa shear stress. The three biopolymer solutions show a gradual increase in oscillation frequency. At a concentration of 1500 mg/L, the storage modulus G′ of all three biopolymers was greater than the loss modulus G″, and the storage modulus G′ and loss modulus G″ of diutan gum were greater than those of xanthan gum and scleroglucan, indicating that the viscoelasticity of the solutions was dominated by the elastic component, and the viscoelasticity of diutan gum was greater than that of xanthan gum and scleroglucan. It can also be seen from Figure 14 that the overlap frequency (the frequency corresponding to the intersection of G′ and G″) of xanthan gum and scleroglucan was considerably larger than that of diutan gum, which shows a typical naturally ordered structure. Another feature of this structure is the strong frequency dependence of the dynamic modulus [2]. Figure 15 shows the relationship between the complex viscosity and the oscillation frequency of the three under the same conditions, and it can be seen that the complex viscosity of diutan gum was also higher than that of xanthan gum and scleroglucan.

At a concentration of 1500 mg/L, diutan gum showed greater viscoelasticity than xanthan gum and scleroglucan, which is due to the formation of a stronger network structure in the diutan gum solution. The molecules of diutan gum possess a distinct double-helix structure, and its side chains are all distributed in the center of the double-helix main chain so that a large number of water molecules can enter into the inner part of the double-helix structure and attach themselves to the side chains through hydrogen bonding. In contrast, the molecules of xanthan gum and scleroglucan solutions are only enhanced by intermolecular and intramolecular forces without a network structure, which is also similar to the findings of Ke Liang et al. [2,29]. Therefore, diutan gum has strong water retention and viscoelastic properties.

### 3.3. Long-Term Stability of Diutan Gum and Scleroglucan

The biopolymers exist in an anaerobic, high-temperature environment during the months-long oil flood in the formation. To test whether biopolymer solutions can maintain sufficient apparent viscosity, aging needs to be simulated in an anaerobic, high-temperature ambient system. A 1500 mg/L solution of diutan gum and scleroglucan with a mineralization of 220 g/L was prepared. The samples were loaded in a glove box after designation and then placed in an oven at 100 °C to carry out the evaluation of the long-term thermal stability performance, and to determine the change in apparent viscosity of the biopolymer solution with aging time.

As can be seen in Figure 16, the apparent viscosity of the diutan gum solution remained relatively stable in the first 10 days of aging, always around 90 mPa·s, but in the period from 10 to 21 days, the apparent viscosity decreased substantially, and on the 21st day, only 13.3 mPa·s remained, corresponding to an 85.01% loss of apparent viscosity. In the period from 21 to 32 days, the apparent viscosity was maintained at around 12 mPa·s. After 32 days, the apparent viscosity dropped to a single digit and gave off a “caramel” odor. The apparent viscosity of the scleroglucan solution did not change much in the first 20 days of aging, remaining at about 70 mPa·s. It then rose slightly from 20 to 39 days, and on the 39th day, the apparent viscosity was still 63.76 mPa·s, corresponding to a loss rate of only 10.46%.

From the above results, it can be determined that the diutan gum has a certain long-term stability in terms of temperature and salt resistance. It can be kept stable for about 10 days at 100 °C and with a 220 g/L mineralization degree, after which, the structure begins to change, the intermolecular hydrogen bonding weakens, and the molecular chain begins to break; thus, the apparent viscosity decreased considerably. On the other hand, the long-term stability of scleroglucan was better than that of diutan gum; it could be maintained at 100 °C and 220 g/L mineralization for about 40 days, which can mainly be attributed to its rigid triple-helix structure.

## 4. Conclusions

In this study, the viscosity changes, and rheological and long-term stability properties of diutan gum, xanthan gum and scleroglucan were investigated under simulated extreme reservoir environments. The results were as follows:(1)At the same temperature and mineralization degree, the thickening and viscosity-increasing performance of diutan gum were better than those of xanthan gum and scleroglucan, and the amount of diutan gum required to achieve the same apparent viscosity was lower than that of xanthan gum and scleroglucan.(2)Diutan gum and scleroglucan showed better temperature resistance than xanthan gum. Diutan gum could withstand a maximum temperature of 100 °C, with an apparent viscosity retention rate of 89%; scleroglucan’s viscosity was stable up to 150 °C, with an apparent viscosity retention rate of 118.27%.(3)Diutan gum and scleroglucan showed better salt resistance than xanthan gum, and both could tolerate up to 220 g/L of mineralized simulated brine, with a 90.12% apparent viscosity retention of diutan gum and 70.04 % of scleroglucan, with minor apparent viscosity loss. The acid and alkali resistance of the three biopolymer solutions was relatively high, with a slight decrease in apparent viscosity under heavily acidic conditions and an approximately constant apparent viscosity under alkaline conditions.(4)In the rheological test, the solutions of diutan gum, xanthan gum, and scleroglucan all distinctly showed pseudoplastic fluid behavior, and the diutan gum had strong water retention and viscoelastic properties due to its peculiar double-helix structure.(5)Both diutan gum and scleroglucan have certain long-term stability in terms of temperature and salt resistance. Diutan gum can remain stable for about 10 days at 100 °C and 220 g/L mineralization, and the apparent viscosity can be maintained at 90 mPa·s. Scleroglucan can remain stable for approximately 40 days in the same reservoir environment, with an apparent viscosity retention rate of 89.54%.

After comprehensively analyzing the temperature, salt, acid and alkali resistance, and long-term stability of the three biopolymers, it is concluded that scleroglucan is more tolerant to the extreme conditions of the reservoir and has the best potential for drive modulation applications.

## Figures and Tables

**Figure 1 polymers-15-04338-f001:**
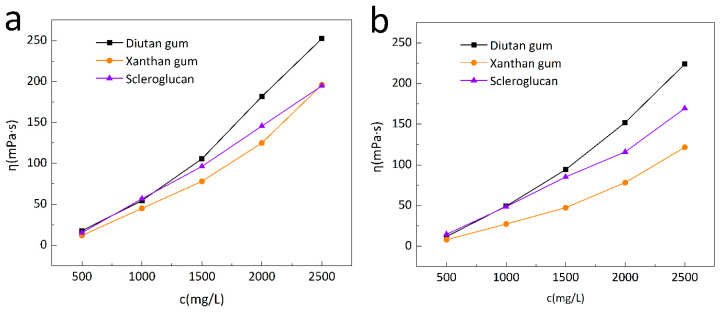
Variation in apparent viscosity of the biopolymer solution with concentration at 20 °C (**a**), and 90 °C (**b**).

**Figure 2 polymers-15-04338-f002:**
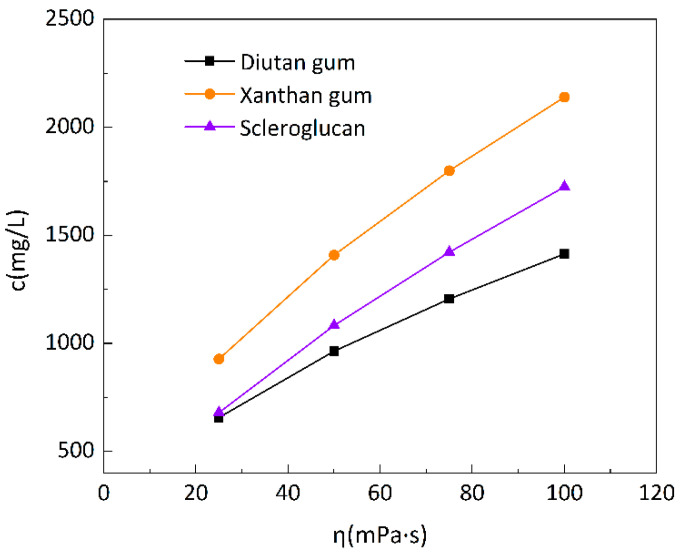
Variation in the biopolymer solution concentration with apparent viscosity at 90 °C.

**Figure 3 polymers-15-04338-f003:**
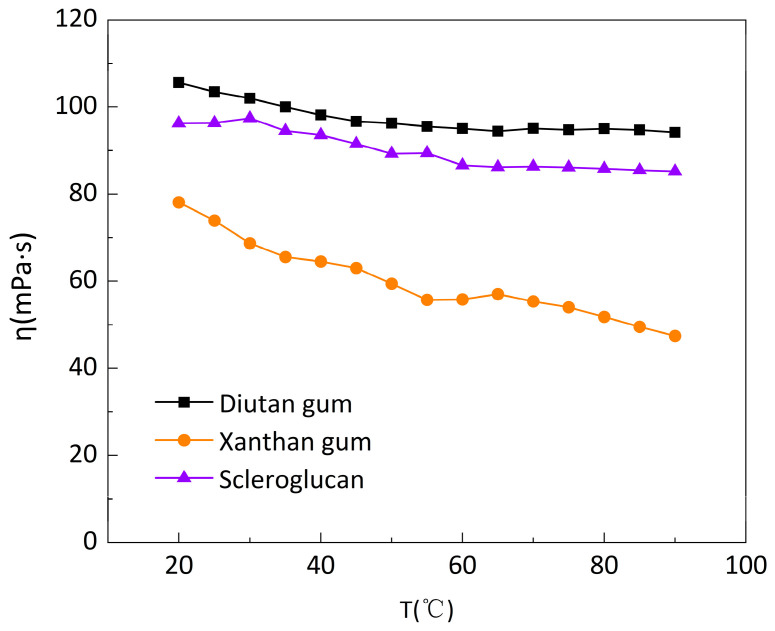
Variation in the apparent viscosity of the biopolymer solutions with temperature at a shear rate of 7.34 s^−1^.

**Figure 4 polymers-15-04338-f004:**
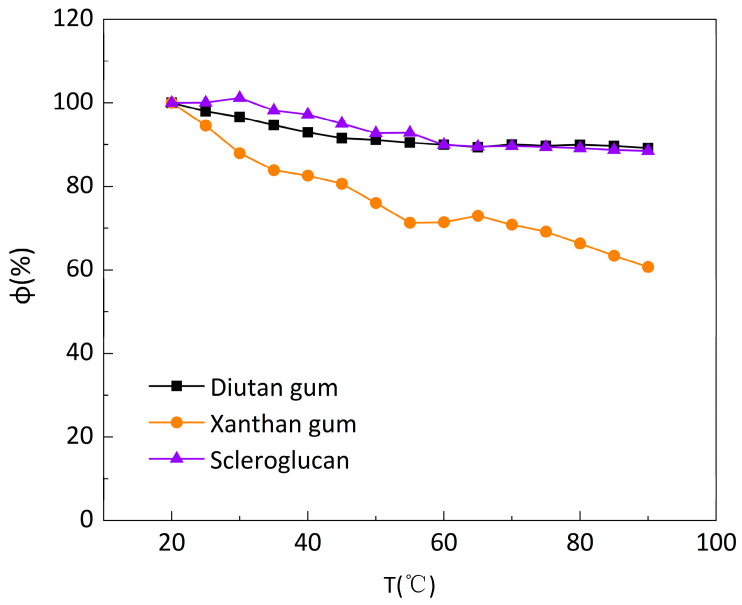
Apparent viscosity retention rate (φ) of biopolymer solutions at different temperatures under a shear rate of 7.34 s^−1^.

**Figure 5 polymers-15-04338-f005:**
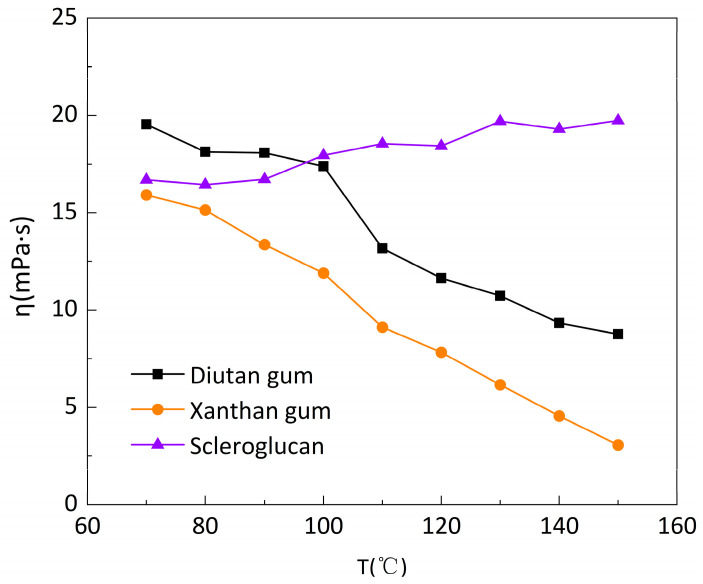
Variation in the apparent viscosity of biopolymer solutions with temperature at a shear rate of 250 s^−1^.

**Figure 6 polymers-15-04338-f006:**
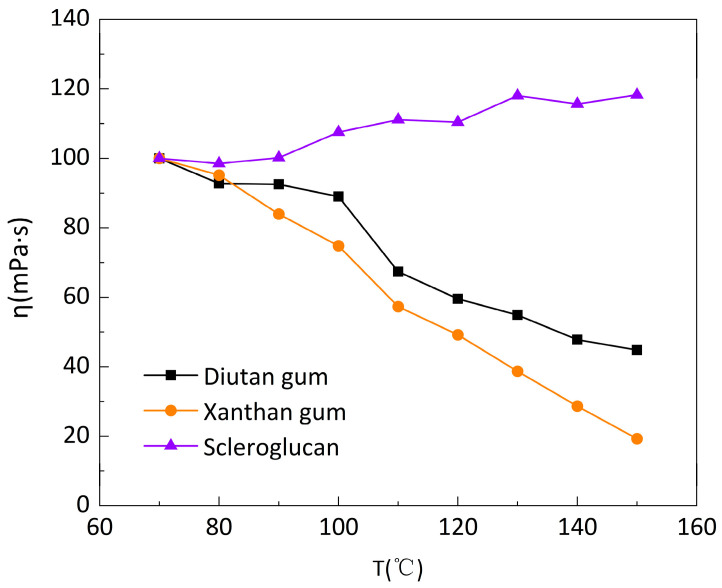
Apparent viscosity retention rate (φ) of biopolymer solutions at different temperatures under a shear rate of 250 s ^−1^.

**Figure 7 polymers-15-04338-f007:**
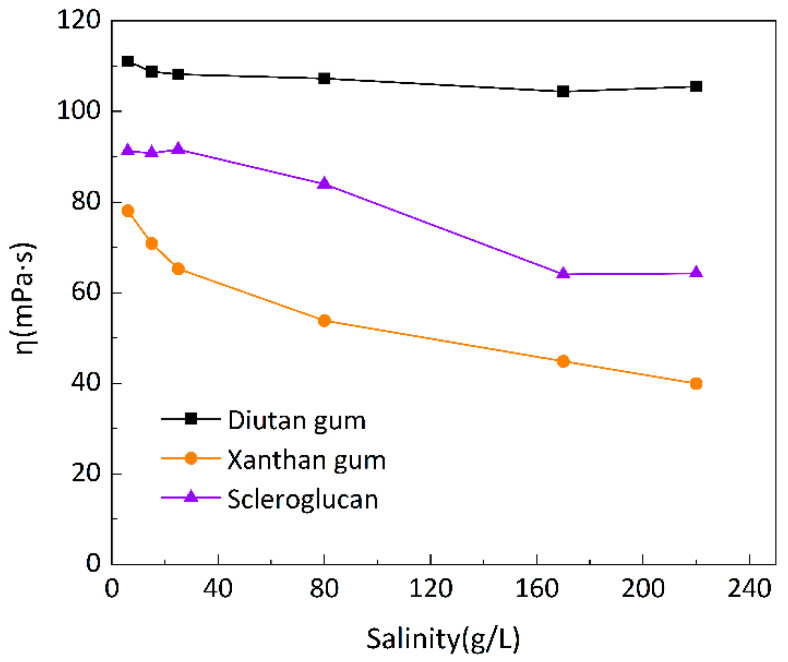
Variation in apparent viscosity of biopolymer solutions as a function of salinity at 20 °C.

**Figure 8 polymers-15-04338-f008:**
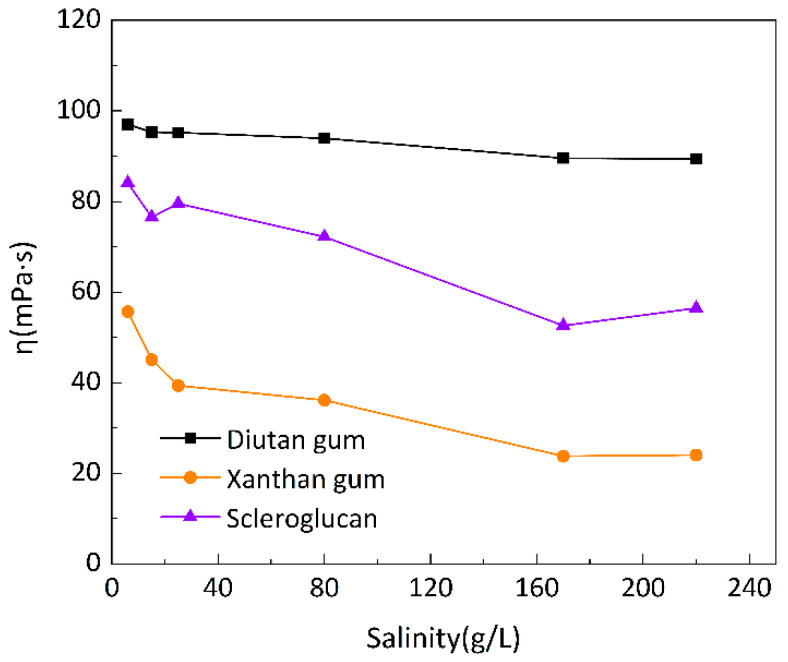
Variation in apparent viscosity of biopolymer solutions as a function of salinity at 55 °C.

**Figure 9 polymers-15-04338-f009:**
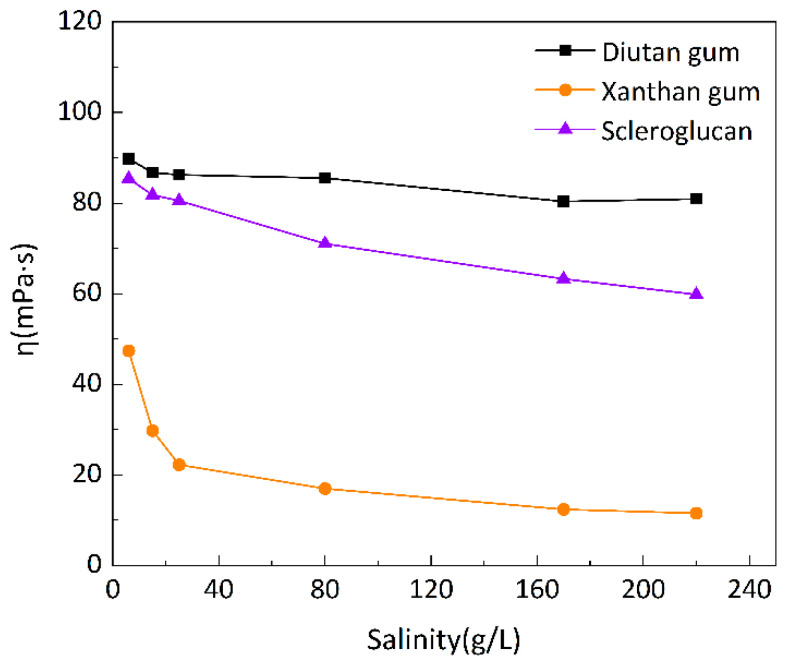
Variation in apparent viscosity of biopolymer solutions as a function of salinity at 90 °C.

**Figure 10 polymers-15-04338-f010:**
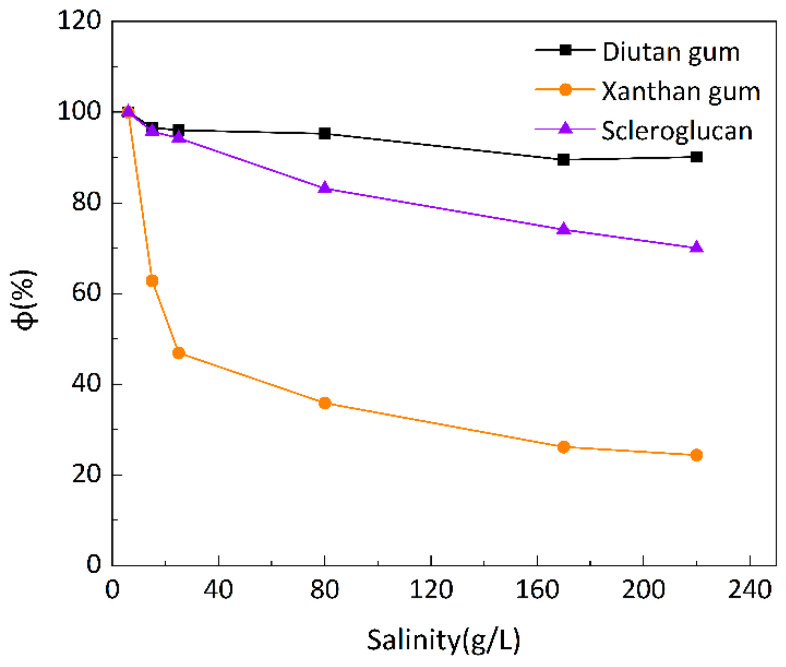
Variation in viscosity retention rate φ of biopolymer solutions as a function of salinity at 90 °C.

**Figure 11 polymers-15-04338-f011:**
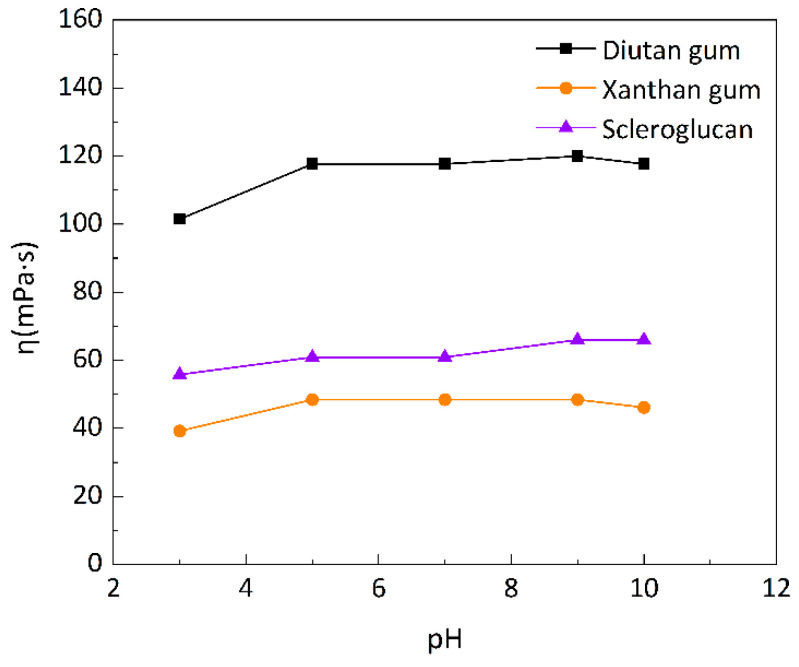
Variation in apparent viscosity of biopolymer solutions with pH at 90 °C.

**Figure 12 polymers-15-04338-f012:**
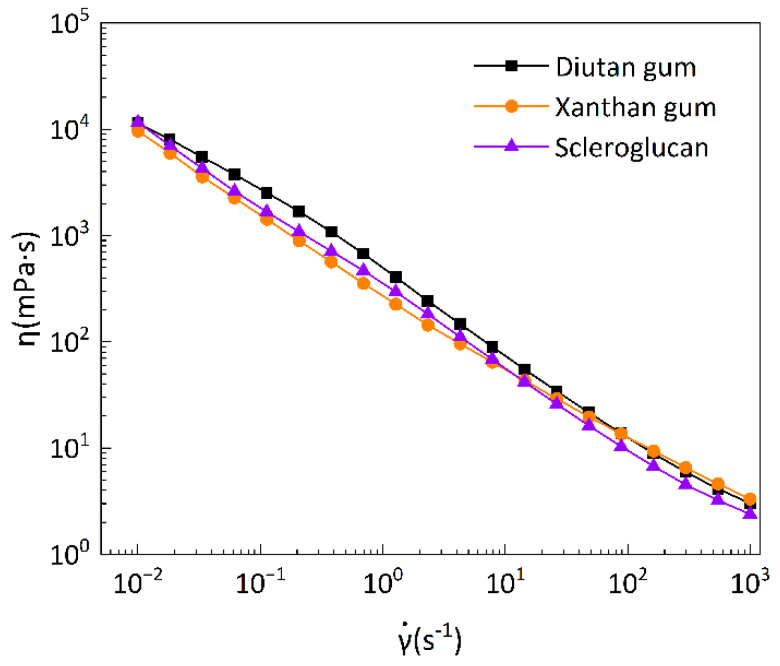
Variation in apparent viscosity of biopolymer solutions with shear rate at 90 °C.

**Figure 13 polymers-15-04338-f013:**
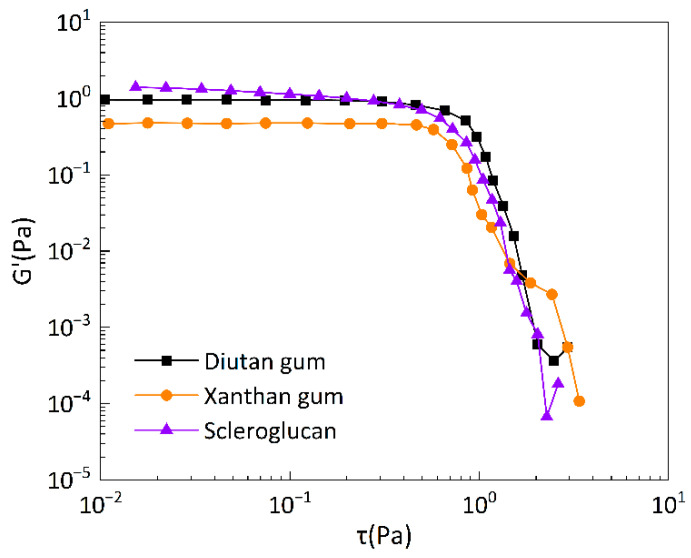
Oscillation amplitude scanning curves of biopolymer solutions at 90 °C and 1 Hz frequency.

**Figure 14 polymers-15-04338-f014:**
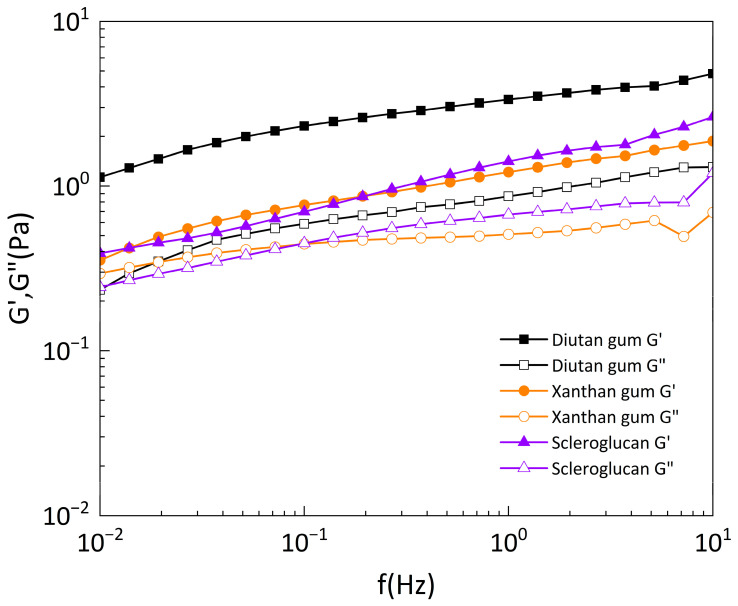
The relationship between the elastic modulus G′ and viscous modulus G″ of the biopolymer solutions and the oscillation frequency at 90 °C and 0.1 Pa shear stress.

**Figure 15 polymers-15-04338-f015:**
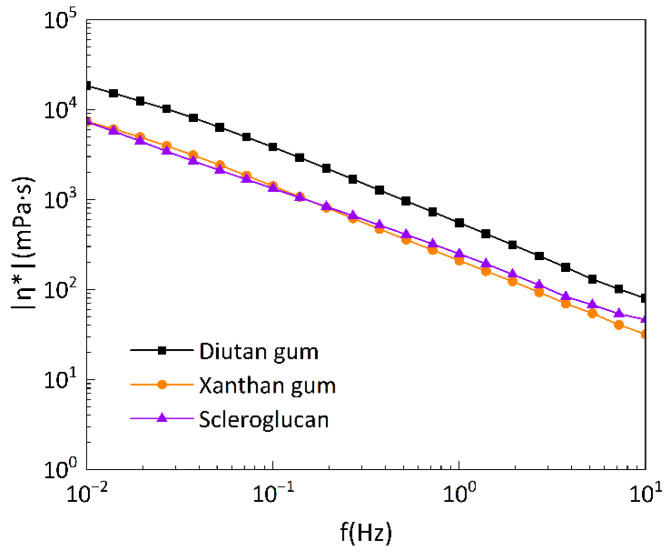
Relationship curves between complex viscosity and oscillation frequency of the biopolymer solutions at 90 °C, under a shear stress of 0.1 Pa.

**Figure 16 polymers-15-04338-f016:**
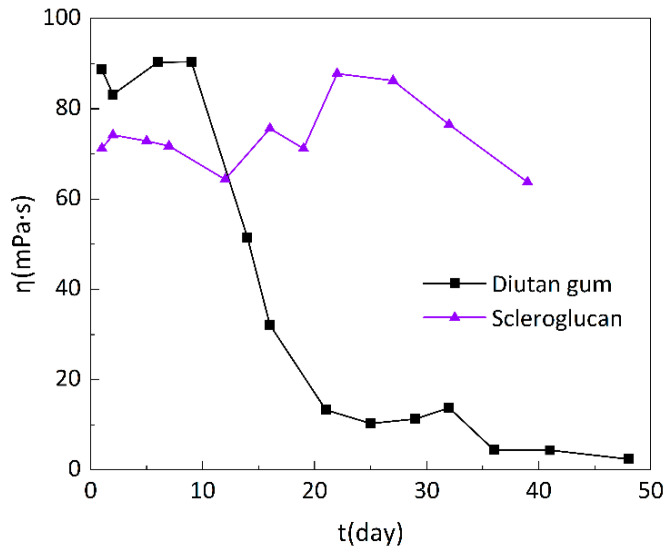
Variation in apparent viscosity of diutan gum and scleroglucan solutions with aging time.

**Table 1 polymers-15-04338-t001:** Simulated brine composition.

Composition (g/L)			Total Mineralization (g/L)
NaCl	CaCl_2_	MgCl_2_·6H_2_O	
192.5	16.5	11	220
148.75	12.75	8.5	170
70	6	4	80
21.875	1.875	1.25	25
13.125	1.125	0.75	15
5.3	0.45	0.3	6

**Table 2 polymers-15-04338-t002:** Power-law model parameters of biopolymer solutions at 90 °C.

Biopolymer Type	K (mPa·s^n^)	n	R^2^	Shear Rate Range (s^−1^)
Diutan gum	444.06	0.250	0.9908	0.01002 <  < 1000
Scleroglucan	334.93	0.248	0.9985	0.01001 <  < 1000
Xanthan gum	186.10	0.305	0.9956	0.01003 <  < 1000

## Data Availability

The raw/processed data required to reproduce these findings cannot be shared at this time, as the data also forms part of an ongoing study.

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
