# Peer review of "Evaluation of Viscosity Changes and Rheological Properties of Diutan Gum, Xanthan Gum, and Scleroglucan in Extreme Reservoirs"

_polymers, 2023, doi:10.3390/polym15214338_

Round 1
Reviewer 1 Report
Comments and Suggestions for Authors
The work submitted for evaluation is a very valuable report on possible application in various industries. the authors planned the preparation of the samples very well and after receiving them, they analyze the rheological properties in various potentially occurring conditions. It analyzes the behavior of the obtained innovative polymer in a multi-factorial manner. The results of the work were presented very well and clearly. The conclusions of the work are a concise summary of the results obtained in the work. I would be fully satisfied if the application description was added even more precisely in the applications.
Author Response
For research article ”A study on viscosity increases and rheological properties of novel biopolymers in extreme reservoirs”
|
Response to Reviewer 1 Comments
|
||
|
1. Summary |
|
|
|
Thank you very much for taking the time to review this manuscript. Please find the detailed responses below and the corresponding revisions in the re-submitted files.
|
||
|
2. Questions for General Evaluation |
Reviewer’s Evaluation |
Response and Revisions |
|
Does the introduction provide sufficient background and include all relevant references? |
Yes/Can be improved/Must be improved/Not applicable |
|
|
Are all the cited references relevant to the research? |
Yes/Can be improved/Must be improved/Not applicable |
|
|
Is the research design appropriate? |
Yes/Can be improved/Must be improved/Not applicable |
|
|
Are the methods adequately described? |
Yes/Can be improved/Must be improved/Not applicable |
|
|
Are the results clearly presented? |
Yes/Can be improved/Must be improved/Not applicable |
|
|
Are the conclusions supported by the results? |
Yes/Can be improved/Must be improved/Not applicable |
|
|
3. Point-by-point response to Comments and Suggestions for Authors |
||
|
Comments: The work submitted for evaluation is a very valuable report on possible application in various industries. the authors planned the preparation of the samples very well and after receiving them, they analyze the rheological properties in various potentially occurring conditions. It analyzes the behavior of the obtained innovative polymer in a multi-factorial manner. The results of the work were presented very well and clearly. The conclusions of the work are a concise summary of the results obtained in the work. I would be fully satisfied if the application description was added even more precisely in the applications.
|
||
|
Response: Thank you for reviewing our article and putting forward valuable comments and suggestions. We are very grateful for your work and feedback, which has greatly helped us in our research. In response to your questions and comments, we have seriously thought and discussed, and made corresponding modifications and improvements. If you have any further questions or suggestions, please feel free to contact us. We look forward to working with you again. |
||

Reviewer 2 Report
Comments and Suggestions for Authors
In the manuscript titled "A Study on Viscosity Enhancement and Rheological Characteristics of Innovative Biopolymers in Challenging Reservoir Environments," the authors have undertaken a rheological analysis of three biopolymers, characterizing them as "innovative." However, it is crucial to acknowledge that the scientific literature is replete with well-established biopolymers, such as Xanthan gum, which have been extensively researched and documented. Even if these particular polymers are indeed novel, a rigorous and comprehensive evaluation is imperative when considering their potential applications in Enhanced Oil Recovery (EOR). Unfortunately, the current work falls short of providing such a comprehensive assessment, focusing primarily on the rheological aspects. Therefore, I regret to inform you that I cannot recommend the publication of this manuscript in the Polymers journal.
Author Response
For research article ” A study on viscosity increases and rheological properties of novel biopolymers in extreme reservoirs”
|
Response to Reviewer 2 Comments
|
||
|
1. Summary |
|
|
|
Thank you very much for taking the time to review this manuscript. Please find the detailed responses below and the corresponding revisions in the re-submitted files.
|
||
|
2. Questions for General Evaluation |
Reviewer’s Evaluation |
Response and Revisions |
|
Does the introduction provide sufficient background and include all relevant references? |
Yes/Can be improved/Must be improved/Not applicable |
|
|
Are all the cited references relevant to the research? |
Yes/Can be improved/Must be improved/Not applicable |
|
|
Is the research design appropriate? |
Yes/Can be improved/Must be improved/Not applicable |
|
|
Are the methods adequately described? |
Yes/Can be improved/Must be improved/Not applicable |
|
|
Are the results clearly presented? |
Yes/Can be improved/Must be improved/Not applicable |
|
|
Are the conclusions supported by the results? |
Yes/Can be improved/Must be improved/Not applicable |
|
|
3. Point-by-point response to Comments and Suggestions for Authors |
||
|
Comments: In the manuscript titled "A Study on Viscosity Enhancement and Rheological Characteristics of Innovative Biopolymers in Challenging Reservoir Environments," the authors have undertaken a rheological analysis of three biopolymers, characterizing them as "innovative." However, it is crucial to acknowledge that the scientific literature is replete with well-established biopolymers, such as Xanthan gum, which have been extensively researched and documented. Even if these particular polymers are indeed novel, a rigorous and comprehensive evaluation is imperative when considering their potential applications in Enhanced Oil Recovery (EOR). Unfortunately, the current work falls short of providing such a comprehensive assessment, focusing primarily on the rheological aspects. Therefore, I regret to inform you that I cannot recommend the publication of this manuscript in the Polymers journal.
|
||
|
Response: Thank you for reviewing our article and putting forward valuable comments and suggestions. We are very grateful for your work and feedback, which has greatly helped us in our research. In response to your questions and comments, we have seriously thought and discussed. In this paper, we will not only evaluate the rheological properties of biopolymer solutions, but also evaluate the viscosification properties of biopolymer solutions under different conditions (high temperature, high salt, acid and base) and long-term (40 days) stability under extreme reservoir conditions. Later, we will also conduct core flooding tests on this. This paper can provide technical support for biopolymer flooding in extreme reservoirs of high temperature and high salt, so we think you can reconsider your suggestion. If you have any further questions or suggestions, please feel free to contact us. We look forward to working with you again.
|
||

Reviewer 3 Report
Comments and Suggestions for Authors
In this paper, the authors compare the viscosity and rheological properties among three different types of polymer polyacrylamide: diutan gum, xanthan gum, and scleroglucan solutions. The experiments are holistic, covering the viscosity-concentration relationship, steady-state rheological shear properties, linear viscoelastic region LVR, and oscillatory frequency scanning. Different extreme conditions were studied, including the variation in temperature, salt concentration, pH, and long-term stability. While I recommend acceptance, revisions are necessary, especially the English must be improved. Many paragraphs are very difficult to follow and understand. Please also answer and address the following comments.
1. In Figure 3, why does the viscosity of scleroglucan increase first then decrease? What does it mean?
2. In line 260 and 261, “diutan gum and sclerogucan are less insentitive to temperature”. Firstly, I think the authors means “less sensitive” instead. Secondly, I cannot agree with the author’s conclusion that diutan gum is stable under temperature. Here is my reason based on the authors’ results:
a. Based on results in Figure 5 and authors' description in line 257, its viscosity decreased over 50 % when increasing the temperature from 70 °C to 150 °C. This amount of decrease is not negligible.
b. Figure 5 and Figure 6 showed from 100 to 110 °C, diutan gum even higher viscosity decreasing slope than xanthan gum.
c. Figure 16 showed poor thermal stability of diutan gum
Please explain this controversy between the conclusion and experimental results.
3. Figure 6’s Y axis is wrong. Should be the apparent viscosity retention rate (φ)
4. Why authors choose 90 °C in the experiments with the variation of pH, steady-state rheological shear test, and oscillation frequency sweep test? Similarly, why choose 100 °C in the long-term stability tests? Please include those information.
5. In lines 347 and 348, the authors explain why xanthan gum has a decrease in viscosity under higher pH. But didn’t explain why diutan gum and scleroglucan are stable. Is it due to diutan gum’s hydron-bonding not being “destructed” (as we inferred from xanthan gum’s explanation)? If so, why?
6. In Figure 13, what does the “plateau” mean for xanthan gum when ꞇ is between 2 and 3 Pa.
7. Can authors provide any scientific supporting information to prove the explanation regarding why diutan gum has poor thermal stability (lines 464-466). It can be either from other’s published work or extra experiments (maybe IR or NMR regarding the hydron bonding?).
8. In Figure 16, why scleroglucan has both the increase and decrease of viscosity as a function of time? Any scientific explanation?
Comments on the Quality of English LanguageGenerally, the English is difficult to follow and understand and must be improved before publication. Some examples:
1. Line 35. Don’t need “The introduction” here.
2. Line 234 and 235, what do you mean “extremely slightly”? How can slightly be extremely?
3. Line 255 and 256, “does not only decrease but also slightly increase”. You can better explain by mentioning “it increases from 20 to 40 C, while starts to decrease after 40 C”.
4. Line 260: should be less “sensitive” I think
5. Line 302: “the apparent viscosity of the xanthan gum solution The apparent viscosity of xan-302 than gum solution” Dual content.
6. Line 303, what do you mean decreased more “definitely”. I don’t think “definitely” is the correct word to describe the extension of decrease.
7. Can you reorganize the paragraph from line 305 to 323. I honestly cannot follow and understand what you want to explain here. Please use some format like “Based on the results, diutan gum has XXX performance in salt resistance, this can be explained By (1) XXX (2) XXX (3) XXX”
8. Line 399: τuntil it reaches
9. Line 405: should be “its critical stress” instead of “the critical stress”.
Author Response
For research article ” A study on viscosity increases and rheological properties of novel biopolymers in extreme reservoirs”
|
Response to Reviewer 3 Comments
|
||
|
1. Summary |
|
|
|
Thank you very much for taking the time to review this manuscript. Please find the detailed responses below and the corresponding revisions in the re-submitted files.
|
||
|
2. Questions for General Evaluation |
Reviewer’s Evaluation |
Response and Revisions |
|
Does the introduction provide sufficient background and include all relevant references? |
Yes/Can be improved/Must be improved/Not applicable |
|
|
Are all the cited references relevant to the research? |
Yes/Can be improved/Must be improved/Not applicable |
|
|
Is the research design appropriate? |
Yes/Can be improved/Must be improved/Not applicable |
|
|
Are the methods adequately described? |
Yes/Can be improved/Must be improved/Not applicable |
|
|
Are the results clearly presented? |
Yes/Can be improved/Must be improved/Not applicable |
|
|
Are the conclusions supported by the results? |
Yes/Can be improved/Must be improved/Not applicable |
|
|
3. Point-by-point response to Comments and Suggestions for Authors |
||
|
Comments 1: In Figure 3, why does the viscosity of scleroglucan increase first then decrease? What does it mean?
|
||
|
Response 1: Thank you for pointing this out. We agree with this comment. Here, we provide an explanation. In Figure 3, the apparent viscosity of scleroglucan solution did increase slightly in the range of 25-30℃, but the amplitude was small, only increasing from 96.35 mPa·s to 97.42 mPa·s, an increase of 1.07 mPa·s. We believe that this is within a reasonable error range, and it can be considered that the apparent viscosity of scleroglucan solution does not change significantly in the range of 25-30℃.
|
||
|
Comments 2: In line 260 and 261, “diutan gum and scleroglucan are less insentitive to temperature”. Firstly, I think the authors means “less sensitive” instead. Secondly, I cannot agree with the author’s conclusion that diutan gum is stable under temperature. Here is my reason based on the authors’ results:
a. Based on results in Figure 5 and authors' description in line 257, its viscosity decreased over 50 % when increasing the temperature from 70 °C to 150 °C. This amount of decrease is not negligible. b. Figure 5 and Figure 6 showed from 100 to 110 °C, diutan gum even higher viscosity decreasing slope than xanthan gum. c. Figure 16 showed poor thermal stability of diutan gum Please explain this controversy between the conclusion and experimental results.
|
||
|
Response 2: Thank you for your revision and comments. It is our opinion that in the evaluation of temperature resistance, the diutan gum and scleroglucan solution are less sensitive to temperature than xanthan gum. The reasons are as follows: (1) It can be seen from figure 3 and figure 4 that the apparent viscosity value and temperature stability of the solution of diutan gum and scleroglucan are always better than that of xanthan gum at 20-70℃. (2) It can be seen from figure 5 and figure 6 that at 100℃, the apparent viscosity of the diutan gum is 17.39mPa·s, and the viscosity retention rate is 89.00%, but at 110℃, the viscosity retention rate drops to 67.40%, which is a large decline, so it can be concluded that the diutan gum can only withstand the high temperature of 100℃. (3) At 100℃ and 110℃, the viscosity retention rate of xanthan gum solution is 74.80% and 57.32%, respectively, and the apparent viscosity value is also low, so the temperature resistance is not as good as that of diutan gum. (4) In the range of 70-150℃, the apparent viscosity of scleroglucan increases with the increase of temperature, which proves that the temperature resistance is excellent. (5) Figure 16 shows the long-term stability evaluation of two biopolymers. The evaluation method is different from the temperature resistance evaluation, so the results are not the same. The following is the test method of the two evaluations. a. Long-term stability evaluation: The two biopolymers were stored in a large number of glass bottles at high temperature without oxygen. A parallel experiment was conducted to take out a glass bottle every few days and quickly measure the apparent viscosity of the solution at 90 ° C, and then plot the apparent visco-time function image. b. Evaluation of temperature resistance: After the solution was configured, the Hakke rheometer was used to measure the change of the apparent viscosity of the biopolymer solution during the heating process within a short time (about 5min for every 10℃ temperature rise).
Comments 3: Figure 6’s Y axis is wrong. Should be the apparent viscosity retention rate (φ)
Response 3: Thank you for your error correction, we have fixed the error. The Y-axis of Figure 6 has been modified to apparent viscosity retention (φ).
Comments 4: Why authors choose 90 °C in the experiments with the variation of pH, steady-state rheological shear test, and oscillation frequency sweep test? Similarly, why choose 100 °C in the long-term stability tests? Please include those information.
Response 4: Thank you for your question. The research content of the text was carried out under simulated extreme reservoir environment, so the experiments with the variation of pH, steady-state rheological shear test, and oscillation frequency sweep test were all carried out under the condition of Hakke rheometer and as high temperature as possible, so 90℃ was chosen. In the long-term stability test, 100℃ was chosen because the temperature of the constant temperature drying box in our laboratory can only support 100℃.
Comments 5: In lines 347 and 348, the authors explain why xanthan gum has a decrease in viscosity under higher pH. But didn’t explain why diutan gum and scleroglucan are stable. Is it due to diutan gum’s hydron-bonding not being “destructed” (as we inferred from xanthan gum’s explanation)? If so, why?
Response 5: Thank you for your question. As the reviewer said, the relative stability of the diutan gum is due to the fact that the strong alkali does not destroy the intramolecular and intermolecular hydrogen bonds of the diutan gum, while the stability of the acid-alkaline resistance of sclerodextran is due to its non-ionic properties and molecular rigidity. There are hydroxyl groups in the glucose unit, and the triple-helix structure is stabilized through the interchain hydrogen bond, and will not change its structure due to the acid-base environment.
Comments 6: In Figure 13, what does the “plateau” mean for xanthan gum when ꞇ is between 2 and 3 Pa.
Response 6: Thank you for your question. The "plateau" discussed in this paper refers to the relatively high platform in the range of τ value less than 1Pa, when the τ value is greater than 1Pa, the molecular aggregates and molecular structure of the polymer have been stretched and deformed, and then we think it is meaningless to discuss. What we want to do in this linear viscoelastic region LVR measurement experiment is to find the critical stress value τc for each biopolymer, so that the subsequent oscillatory frequency sweep test can obtain the storage modulus G'and the loss modulus G" of the polymer solution.
Comments 7: Can authors provide any scientific supporting information to prove the explanation regarding why diutan gum has poor thermal stability (lines 464-466). It can be either from other’s published work or extra experiments (maybe IR or NMR regarding the hydron bonding?).
Response 7: Thank you for your question and advice. In this paper, the long-term stability of diutan gum and scleroglucan solution was studied under the following conditions: the total salinity was 220g/L and the temperature was 100℃. In the literature and materials the authors has read so far, we can not find such a high degree of salinity and temperature conditions coexist. For example, when doing a long-term stability study of diutan gum or scleroglucan, the conditions applied by Yajun Li are: total salinity 9.5g/L, temperature 75℃; The conditions applied by Ke Liang are as follows: total salinity 100.1g/L, temperature 85℃; The conditions imposed by Hongtao Zhou are: total salinity 0g/L, temperature 60℃.According to Long Xu's research, the molecular structure of xanthan gum and diutan gum is similar, both of which have double helix structure. Under the condition of total salinity of 55.5g/L and temperature of 75℃, diutan gum can maintain its conformation well, while the backbone to double helix of xanthan gum is gradually loosened. The side chain in double helix also gradually disconnects, eventually taking on the status of disordered coils. From this, we made a reasonable assumption. Compared with Long Xu's study, when the total salinity is nearly four times higher and the temperature is 25℃ higher, the molecular structure of the diutan gum also changes similar to the above xanthan gum molecular structure, and the double helix structure no longer exists, resulting in the apparent viscosity stability of the diutan gum can only be maintained for about 10 days.
Comments 8: In Figure 16, why scleroglucan has both the increase and decrease of viscosity as a function of time? Any scientific explanation?
Response 8: Thank you for your question. The reason for the slight increase in apparent viscosity of scleroglucan solution is that this is due to the fact that the dissolution of biopolymer molecules in water involves three processes: hydration, swelling and dissolution. Due to the large molecular weight of biopolymers and the common polymerization during the dissolution process, resulting in delayed hydration, the viscosity of biopolymer solution will increase slightly when placed in a short period of time. The reason for the late decrease in the apparent viscosity of scleroglucan is that the triple-helix structure of scleroglucan begins to undergo hydrolysis under a long-term high temperature environment, and the internal force of the molecule is not enough to maintain the external pressure, resulting in the late decrease in apparent viscosity.
|
||
|
4. Response to Comments on the Quality of English Language Thank you for pointing out the errors in the English language, we have used MDPI's language editing services to polish and correct the article. |
||

Round 2
Reviewer 2 Report
Comments and Suggestions for Authors
Still author has not added core flood results and they are considering it for future work. However, the work covers the rheological evaluation.
Author Response
Thank you for reviewing our article and putting forward valuable comments and suggestions. We are very grateful for your work and feedback, which has greatly helped us in our research. In response to your questions and comments, we have seriously thought and discussed, and made corresponding modifications and improvements.
If you have any further questions or suggestions, please feel free to contact us.
We look forward to working with you again.
